# Combination of Nanodelivery Systems and Constituents Derived from Novel Foods: A Comprehensive Review

**DOI:** 10.3390/pharmaceutics15112614

**Published:** 2023-11-11

**Authors:** Eleonora Truzzi, Davide Bertelli, Anna Rita Bilia, Giulia Vanti, Eleonora Maretti, Eliana Leo

**Affiliations:** 1Department of Chemical and Geological Sciences, University of Modena and Reggio Emilia, Via G. Campi 103, 41125 Modena, Italy; eleonora.truzzi@unimore.it; 2Department of Life Sciences, University of Modena and Reggio Emilia, Via G. Campi 103, 41125 Modena, Italy; davide.bertelli@unimore.it; 3Department of Chemistry “Ugo Schiff” (DICUS), University of Florence, Via Ugo Schiff 6, 50019 Sesto Fiorentino, Italy; ar.bilia@unifi.it (A.R.B.); giulia.vanti@unifi.it (G.V.)

**Keywords:** novel food, health compounds, natural polymers, nanoparticles, nanocarriers, nanoformulations, nanoceuticals, EFSA, nanotechnology

## Abstract

Novel Food is a new category of food, regulated by the European Union Directive No. 2015/2283. This latter norm defines a food as “Novel” if it was not used “for human consumption to a significant degree within the Union before the date of entry into force of that regulation, namely 15 May 1997”. Recently, Novel Foods have received increased interest from researchers worldwide. In this sense, the key areas of interest are the discovery of new benefits for human health and the exploitation of these novel sources of materials in new fields of application. An emerging area in the pharmaceutical and medicinal fields is nanotechnology, which deals with the development of new delivery systems at a nanometric scale. In this context, this review aims to summarize the recent advances on the design and characterization of nanodelivery systems based on materials belonging to the Novel Food list, as well as on nanoceutical products formulated for delivering compounds derived from Novel Foods. Additionally, the safety hazard of using nanoparticles in food products, i.e., food supplements, has been discussed in view of the current European regulation, which considers nanomaterials as Novel Foods.

## 1. Introduction

According to the United Nations, the world’s population is estimated to reach 9.7 billion in 2050. Population growth is induced by the decreasing mortality rate and an average fertility still at high levels. The demographic increase will not be homogenously projected worldwide. Indeed, the global population increment is occurring mainly in Asian and African countries, where people are the most affected by hunger and the impossibility of following a balanced healthy diet [1]. This situation was exacerbated during the global COVID-19 pandemic period, and it is expected to tragically worsen. In 2009, the dramatic foreseen scenario alerted the Food and Agriculture Organization of the United Nations (FAO), which demanded a 70% increment in the global food production by 2050 in a sustainable manner [2]. In this context, innovation is essential for countering the drastic challenge that the food system is facing, and technological and scientific progress are precious allies. Novel Food represents an example of the fundamental role of innovation in guaranteeing food security. Indeed, Novel Food is a new category of foods regulated by the European Union. The first definition of Novel Food was laid down in the Regulation of the European Commission (EC) No. 258/97, which was repealed by Regulation (EU) 2015/2283. This latter norm defines a food as “Novel” if it was not used “for human consumption to a significant degree within the Union before the date of entry into force of that Regulation, namely 15 May 1997”. The EU regulation was then implemented by the Regulation (EU) 2017/2470, which provides a detailed list of Novel Foods [3]. This list is continuously updated by the Commission Implementing Regulations (EU). The list includes “foods derived from plants and animals, produced by non-traditional breeding techniques, and foods modified by new production processes, such as nanotechnology and nanoscience, which might have an impact on food”. Novel Foods are authorized by the European Food Safety Authority (EFSA) only if all the criteria laid down in Regulation (EU) 2015/2283 are fulfilled after performing a process of risk assessment. Specifically, Novel Foods should not be dangerous or misleading to consumers, and not be so different from the foods or food ingredients they are intended to replace (e.g., non-traditional production methods). Therefore, a Novel Food is after all considered as food for all intents and purposes. The denomination “Novel Food”, intended as “foods that have not been used for human consumption to a significant degree within the EU before 15 May 1997”, does not exist in many jurisdictions outside the European Union. As an example, the U.S. Food and Drug Administration (FDA) agency does not provide any formal definition or category for new foods. Conversely, any new food is regulated as any other food via the food additive process if it is not considered Generally Recognized as Safe (GRAS) or meets an exception under the Federal Food, Drug, and Cosmetic Act [4].

The authorization process of Novel Foods was modified by Regulation (EU) 2018/456 to smooth and accelerate the approval. Indeed, unlike previously, the authorization is made by a centralized system. The EC analyzes each application to verify that all criteria are respected. Then, the application is submitted to EFSA for a scientific risk assessment instead of each EU member country. In this way, the entire process lasts for not more than nine months. This procedure is further sped up in the case of traditional food from a third country. In this instance, the submission of a notification to the EC is enough. After evaluating its validity, the EC has to forward the notification to the EFSA and member states and, if doubts about the safety are not raised, after four months, the traditional food is authorized to be placed on the market. A summary of this procedure is displayed in Figure 1.

Starting from the publication of the first Novel Food list in 2017, the rate of consumption of Novel Food has rapidly grown, supplying new sources of lipids, carbohydrates, proteins, fibers, and other nutrients with beneficial effects on human health. These classes of compounds have received increased interest from researchers worldwide. In this sense, the key areas of interest were the discovery of new benefits for human health and the exploitation of these novel sources of materials in new fields of application, such as pharmaceutical technology. Pharmaceutical technology is one of the pharmaceutical sciences involving the formulation, preparation, and composition of commercially manufactured drugs. An emerging area in the pharmaceutical and medicinal fields is nanotechnology, which deals with developing, amongst others, new delivery systems at a nanometric (1 nm–100 nm) and micrometric (100 nm–100 μm) range (10 nm–1000 nm) [5]. Several nanocarriers have been approved for drug delivery vehicles in human applications, exhibiting a wide range of compositions and structures. In particular, this review focuses primarily on the prevalent polymeric- and lipid-based nanoparticles.

Nanocarriers were studied in depth and exploited in recent decades to increase the bioavailability and solubility of bioactive compounds, along with the improvement of their activity. Additionally, nanoparticles are capable of protecting physico-chemically labile bioactive compounds, providing them with an extended stability during storage or under physiological conditions. Nanotechnology has been applied both to deliver compounds with health properties and to produce nanoparticles using food-derived materials, also called “nanoceutical materials”, for embedding compounds with beneficial activities on human health. Consumers’ increasing interest and trust in natural health products have boosted this scientific research in recent decades, as previously reported [6]. Nutraceuticals are essential nutrients for human health and well-being and are present in edible food products. Numerous in vitro studies have proved that many nutraceuticals are multifunctional compounds with healthy features comparable to those of conventional drugs. For this reason, nutraceuticals are considered “pharmaceutical-grade compounds”.

Due to their natural source, Novel Food can represent another new and not fully exploited reservoir of nutraceuticals. Indeed, Novel Food can also be considered an important source of raw matter to produce a nano-drug delivery system. The application of Novel Foods as new nanoceutical materials or bioactive compounds embedded into nanocarriers is a new research field, as these sources have been employed only in recent decades (Figure 2). Thus, this review aims to summarize the recent advances in the production of nanocarrier systems for delivering Novel Foods and nanocarriers based on nanoceutical materials belonging to the Novel Food list. This review considers the new advances from 2018 to the present, focusing on “nano-” strategies for oral administration.

## 2. Novel Food Materials for Nanoparticle Production

The main advantages of using food-derived materials for the fabrication of nanoparticles are the extremely high biocompatibility, safety, and respect of the food-grade requisites, such as the employment of solvents that comply with food-grade standards. This latter aspect is essential for the potential application of nanoparticles in food supplements or fortified foods with an increased nutritional value. In this section, only materials extracted from Novel Foods were considered, and their advantages were underlined.

### 2.1. Polymeric Nanoparticles

Nanoparticle-mediated smart delivery systems (np-DS) can exhibit favorable multifunctional characteristics for the delivery of nutrients or bioactive molecules for successful targeting. In this scenario, the np-DS application is evolving from traditional raw food products to advanced technologies in novel food engineering, which ensure purity and functionality.

Polymeric particles (PNp), intended as nano- and microparticles, are an important class of drug delivery system for the suitable delivery of bioactive compounds. PNp are defined as particles of diameter <1 μm composed of either biodegradable or non-biodegradable biopolymers that have been recently reviewed by M. Elmowafy et al. [7]. The most recent source of polymers for PNp synthesis are Novel Foods, as summarized in Table 1.

Chitosan is the most applied polymer from Novel Food used in PNp assembling. Chitosan is derived from chitin, a naturally occurring polysaccharide found in the exoskeletons of crabs, lobsters, shrimp, and insects. Chitosan is essentially the N-deacetylated form of chitin and is composed of 2-acetamido-2-deoxy-β-d-glucose units linked together through β (1→4) linkage. Chitin, like cellulose, serves as a structural polysaccharide in these organisms, but it differs from cellulose in various ways. Chitin is highly hydrophobic and does not dissolve in water or most organic solvents. However, when chitin undergoes deacetylation, it transforms into chitosan, which is soluble in dilute acids, such as acetic and formic acids. Chitosan possesses several advantageous characteristics, including being non-toxic, biocompatible, and biodegradable [54]. To produce 1 kg of chitosan with a 70% deacetylation level from shrimp shells, you would need 6.3 kg of hydrochloric acid (HCl) and 1.8 kg of sodium hydroxide (NaOH). This process also requires nitrogen, processed water (0.5 t), and cooling water (0.9 t). When estimating the production cost of chitosan, it is important to consider various factors, such as transportation costs, which can vary depending on labor expenses and the location of the production facility. These costs can have a significant impact on the overall cost of chitosan production [55]. Different varieties of fungi are selected as a source of chitosan among the novel foods. Fungi are organisms that contain chitin as a major structural component in their cell walls [56]. For example, *Absidia coerulea* and *Aspergillus niger* were found to have a high proportion of chitin and chitosan [57,58]. Moreover, chitosan can be prepared using mycelia waste or mushroom industry waste. The fungal waste biomass is of growing importance, and is generated by biotech industries and the fermentation of zygomycete mushrooms. As a matter of fact, thousands of tons of waste fungal biomass are produced by Mycotech industries every year and are landfilled or incinerated for easy disposal. The use of mushrooms as a source of chitosan is advantageous due to the homogeneity and highly deacetylated nature of the product [56]. Mane and co-workers examined fungal organisms in the waste fungal biomass of the wine and mushroom industries and the fermentation of the zygomycete fungus *B. poitrasii* [12]. In addition to the production and characterization of the extracted chitosan, the synthesis of fungal chitosan nanoparticles was also conducted to determine their antifungal potential against human pathogenic fungi. Chitosan obtained from ascomycete yeast and basidiomycete fungi has a lower molecular weight when compared to chitosan from *B. poitrasii*. Additionally, it exhibits poor antifungal properties, which can be attributed to its very low molecular weight and low polycationic nature, meaning it has a low percentage of deacetylation when compared to the chitosan from *B. poitrasii*. Even though the raw waste biomass available for ascomycetes and basidiomycetes fungi may be more abundant, the chitosan extracted from them cannot be effectively used for sanitary applications due to its low cationic nature, which affects its antifungal properties. The authors suggest that % deacetylation is more important than molecular weight for the antifungal activity of chitosan polymers and their nanoparticles. In contrast, chitosan isolated from the zygomycete fungus *B. poitrasii*, with a deacetylation level of 92.78% and a molecular weight of 46.33 kDa, appears to be a promising source for healthcare applications due to its higher cationic nature and better antifungal potential. Nanoparticles produced using chitosan from *B. poitrasii* have demonstrated effective inhibition of the growth of human fungal pathogens at a lower minimum inhibitory concentration (MIC90) when compared to the traditional chitosan polymer. These findings suggest that chitosan nanoparticles prepared from *B. poitrasii* chitosan, particularly when it has a high deacetylation level, have the potential to be a powerful and safe natural antifungal agent.

Wu et al. demonstrated that fungal chitosan from the mycelia of *Agaricus bisporus* (Lange) Sing Chaidam was similar to commercial crab chitosan [13]. In this context, chitosan derived from fungi with a lower molecular weight was employed to coat modified betulinic acid-loaded liposomes. When compared to the commercially available crab chitosan, these fungal chitosan-coated liposomes exhibit several advantages, including smaller particle size, increased stability, and enhanced encapsulation efficiency. The zeta potential and encapsulation efficiency of these liposomes depend on the concentration of the chitosan used in the coating process.

Another important polysaccharide is the guar gum obtained from the ground endosperm of the seed of natural strains of guar *Cyamopsis tetragonolobus* L. Taub. (*Leguminosae* family). Guar gum is a polymer capable of creating a hydrocolloid system in the presence of water and is resistant to the conditions of the gastrointestinal tract, such as acidic pH and microorganisms. Furthermore, guar displays beneficial properties, such as the control of diabetes, and anti-inflammatory and antimicrobial activities. Nanoformulations based on guar gum are able to protect the embedded compound from the stomach to the colon, to treat diseases associated with the last tract of the intestine [27].

Starting with *Astragalus membranaceus*, a mushroom used in traditional Chinese medicine for more than 2000 years to strengthen human immunity and cardiovascular disorders [59,60], polysaccharides named *Astragalus* polysaccharides (APS) were extracted with water from *Astragalus* roots [61]. These are some of the main active components of *Astragalus membranaceus* and are listed as Novel Food. APS has potent anti-inflammatory, organ protective, anti-insulin resistant, and anti-tumor activities and consists of rhamnose, arabinose, and glucose [62,63]. To improve their activity, Meng et al. formulated APS as nanoparticles by cross-linking them with sodium selenite via simple preparation conditions [8]. The bioactivities of selenium–aminopolysaccharide (Se-APS) were investigated in vitro. The results demonstrated that APS-selenide nanoparticles have the potential to not only enhance the proliferation of T-lymphocytes but also effectively inhibit the malignant proliferation of HepG2 cells. Moreover, they were found to reduce cell migration and invasion. These findings suggest that Se-APS nanoparticles hold promise as a novel supplement due to their high selenium content and strong bioactivity. Similarly, nanoparticles composed of starch from *Digitaria exilis* were obtained via cross-linking with multifunctional excipients, such as citric acid and sodium tripolyphosphate. These nanoparticles were designed for the delivery of ibuprofen [20].

Among novel and organic-based polysaccharide biopolymers, chondroitin sulphate (CS)—from edible marine (such as crustaceans, mollusks, and Echinodermata) and animal skin and cartilage—captured increasing attention for various biomedical applications, particularly as the main component of np-DS [14,15]. CS is also present in several food supplements as an alternative medicine to treat osteoarthritis; however, being extracted from natural sources, its chemical composition and biological efficacy might extremely vary. An alternative method for the production of CS was proposed and involves the chemical sulphation of chondroitin derived from fermentation with the bacterium *Escherichia coli*. This type of CS is included in the Novel Food list. CS is a negatively charged heteropolysaccharide belonging to the family of sulfated glycosaminoglycans (GAGs). It is found in a wide range of organisms, including vertebrates, invertebrates, and bacteria. The length of the CS chain can vary depending on its source of origin. For instance, tracheal chondroitin sulfate typically has a molecular weight in the range of 20–25 kDa, while shark chondroitin sulfate tends to have a higher molecular weight, typically falling within the range of 50–80 kDa [16]. CS is valued for its various therapeutic properties, which include anti-inflammatory, antioxidant, and anti-thrombotic effects. These effects are primarily attributed to its ability to reduce the levels of interleukin-1β (IL-1β), nuclear factor- κβ (NF-κβ), and matrix-degrading enzymes [64]. As a result of these properties, CS is utilized as both a nutritional supplement and an over-the-counter drug for the treatment of osteoarthritis. Furthermore, CS possesses versatile characteristics that make it an ideal biomaterial for the development of np-DS. These systems can be used for the controlled release of anti-cancer and anti-inflammatory drugs, antigens, and protein/peptide molecules. This potential for a sustained release has been reviewed in a recent work by Sharma and colleagues [14]. Given its high hydrodynamic volume and hydrophilicity, CS can be used to inhibit unwanted interactions with plasma proteins and cells during circulation in the body of nanoparticle systems. In addition, CS has been used in the development of nanocarriers to improve targeted drug delivery for cancer and inflammatory diseases due to its interaction with the CD44 receptor or to achieve an intracellular targeting of drugs [15].

Cellulose derivatives are other polymers commonly used for nanoparticle production. A natural source of cellulose inserted in the Novel Food list is the sugar bagasse fiber. Sugarcane bagasse is the dry pulpy residue left after the extraction of juice from sugarcane. In light of a circular economy, the recovery of fibers from this waste is increasing. Sugarcane bagasse is mainly composed of cellulose and hemicellulose, which represent 70–75% and 20–25% of its mass, respectively, in accordance with the EU Directive No. 2015/2283. Recently, derivatives of these polysaccharides were proposed. As an example, Kumar H. et al. developed copper oxide nanocomposite (CuO-NP) hydrogels based on lab-made carboxymethylcellulose (CMC) from sugarcane bagasse. CMC is a class of hydrophilic polymers, which, in contrast to cellulose, are easily resolved in water and have a perfect swelling strength. These CuO-NPs demonstrated a greater antibacterial activity against Gram-positive and Gram-negative bacteria compared to synthetic CMC purchased from the market. This evidence indicates that the hydrogels developed from sugarcane bagasse have a great potential application in various medical areas, for instance, in drug delivery and wound healing [48].

Fucoidan (FUC), a marine polysaccharide derived from the cell wall of marine algae, is considered a highly promising polymer for pharmaceutical research and the creation of an innovative drug delivery system [24]. FUC is a non-toxic, biodegradable, and biocompatible polymer that has received approval from the U.S. FDA as a food ingredient, specifically categorized as “Generally Recognized as Safe.” Fucoidan possesses a range of beneficial properties, including its role as an emulsifier and viscosity enhancer. The primary sources of FUC are marine brown algae, including members of the Phaeophyta, Laminariaceae, Fucaceae, Chordariaceae, and Alariaceae families [26], as well as sea cucumbers and sea urchin eggs, and some seagrasses (e.g., Cymodoceaceae) [65]. The marine polysaccharide FUC exhibits unique structural and chemical characteristics due to the presence of the -OSO3H group, which imparts strong acidic properties to it. This feature makes the FUC capable of forming polyelectrolyte complexes (PECs) with different polycations, of different natures, for the development of safe and effective delivery systems. Among these, modified (prolonged, controlled, and targeted) drug release systems, such as polyelectrolyte particles, core–shell NPs, multilayer NPs, and self-assembling polymer structures with stable and reproducible properties from FUC, have been recently reviewed [24].

Phytoglycogen (PG) is a highly branched polymer of glucose (a type of starch-like carbohydrate) which is produced as soft, compact nanoparticles. Indeed, when PG is introduced into water, it assumes the form of monodisperse, low-viscosity, and extensively hydrated colloidal particles with nanosized dimensions ranging from 30 to 110 nanometers [66]. The primary origin of PG can be traced to the sugary-1 mutant variant of maize kernels, which is a prominent genetic makeup of sweet corn. However, PG has also been detected in various other widespread plant species, such as sorghum and rice. The Novel Food Directive No. 2017/2470 allows the extraction of PG from non-genetic modified sweet corn via conventional food processing techniques. The distinctive attributes of PG, including its dendrimeric arrangement, as well as its non-toxicity and biodegradability, render phytoglycogen nanoparticles exceptionally suitable for applications in the pharmaceutical, cosmeceutical, or nutraceutical areas. Indeed, owing to its remarkable characteristics related to dispersibility, structure, and morphology, PG has been recently used in its native form or after undergoing chemical modifications in the form of nanoparticles to enhance the solubility of bioactive compounds that are poorly soluble in water [32,33]. Thanks to the high concentration of hydroxyl groups on the surface of PG nanoparticles, they are well suited for a wide array of chemical alterations [34].

Yuhao Chen et al. studied the molecular interactions between modified PG nanoparticles, i.e., carboxymethyl phytoglycogen (CMPG) and casein (NaCas) nanoparticles under different complexation conditions. Their interaction was studied to obtain a polyelectrolyte nanocomplex via an electrostatic interaction for the encapsulation and delivery of curcumin, employed as a hydrophobic bioactive compound model. Electrostatic interactions and hydrogen bonding were the driving forces that enabled a nanocomplex to be obtained, where the hydrophobic interactions had a restricted influence. A nanocomplex, approximately 65 nm in size with a PDI of 0.2, was achieved through fine-tuned preparation parameters. It was conjectured that the heating process significantly contributed to the nanocomplex formation by promoting partial aggregation of NaCas, yet it had an adverse impact on curcumin’s encapsulation efficiency, reducing it by approximately 40%. This outcome indicated the need to explore alternative chemical modifications of native PG for the creation of nanocomplexes with improved encapsulation and delivery capabilities [35]. Octenyl succinic anhydride (OSA) was employed as a modifier, and it is a widely used food-grade starch modification to achieve emulsification and encapsulation in various food and non-food applications [37]. Simmons and co-workers studied how the chemical substitution of the hydroxyl groups of PG with OSA influenced the nanoparticle shape and surface. The detailed analysis with small-angle neutron scattering (SANS) demonstrated that the degree of the substitution of PG nanoparticles with OSA has a central role in particle structure. Specifically, this study has provided evidence that native PG nanoparticles with a core of radius 210 Å (21 nm) are decorated by hairy chains, and by increasing the degree of substitution the hair chains collapse to form more compact particles with small seeds on the surface. This information has important implications for understanding how chemical modifications influence the physical properties of PG nanoparticles, which might be suitable for a wide variety of scientific and industrial applications [32]. By examining the interactions of PG nanoparticles with several representative proteins, Yingshan Ma et al. found that bovine serum albumin (BSA), analogous to the most abundant protein in human plasma, adsorbs to PG via hydrogen bonding. Under ambient conditions, the BSA layer originally endowed electroneutral PG nanoparticles with additional ligand-binding capabilities, enabling the interactions with and the further attachment of active pharmaceutical ingredients. In particular, they showed the attachment of methotrexate (MTX) to PG nanoparticles and BSA nanoconjugates [36].

Finally, several charged polysaccharides were proposed as gene delivery vectors for tumor immune-targeted therapy loaded with bioactive compounds for the production of multifunctional nanoparticles. β-glucans isolated from yeast *Saccharomyces cerevisiae* and polysaccharides from *Panax notoginseng* are examples included in the Novel Food list [31,52,53].

The preparation methods of polymer nanodeliveries are very different depending on the nature of the starting polymer. In the case of polysaccharides, ionic gelation methods are the most used [10,11,14], while electrospinning is used frequently for the preparation of nanofibers [12,13]. On the other hand, naturally occurring nanoparticles, such as PC nanoparticles, are extracted from vegetable sources [32,34].

### 2.2. Lipid-Based Nanoparticles

Lipid-based nanoparticles are a class of nanocarriers composed of lipids with different characteristics depending on the purpose and the active ingredient to be transported. Among this class, the most used delivery systems are liposomes, solid lipid nanoparticles (SLNs), and nanostructured lipid nanoparticles (NLCs).

Liposomes are vesical nanoparticles able to load both hydrophilic and lipophilic compounds. Their spherical 3D structure is composed of an internal aqueous core surrounded by phospholipid bilayers usually stabilized by cholesterol. The use of phytosterols in the place of cholesterol for stabilizing liposome membranes is widely investigated to avoid its negative effects, such as the increased risk of cardiovascular disease. Phytosterols are naturally occurring lipids extracted from natural sources (such as rice, vegetable oils, and soybeans) included in the Novel Food list. They are expected to exhibit the same unique features of cholesterol due to their close chemical structure.

A comparison was made by Jovanovic et al., where liposomes with cholesterol and β-sitosterol were prepared and characterized. The use of phytosterol allowed smaller liposomes to be obtained with an increased surface charge and a higher stability of the system compared to liposomes with cholesterol [39].

As mentioned previously, phospholipids are the main component of liposomes and their origin can differ (e.g., soybean, egg yolk, or milk). In the field of Novel Food, egg yolk and chia seed lipids are the principal sources. Also, phosphatidylserine isolated from soya and fish phospholipids are included. Many articles in the literature reported the use of phospholipids from egg yolks to formulate liposomes [40,41,42]. The various composition and origin of egg yolk extract allows us to obtain liposomes with different characteristics and properties in terms of stability, permeability, and morphology [42,67,68].

Meanwhile, Kuznetcova et al. decided to use a phospholipid-rich solid fraction of chia seed lipids to prepare nanoliposomes. They obtained carriers of about 120 nm with a negative surface charge [11].

Considering the limitation of liposomes, both in the technological field and in the oral administration route, the addition of excipients capable of protecting them from freeze drying and the extreme environment of the gastrointestinal tract have to be taken into consideration. Chen et al., in order to deliver bioflavonoids via the oral route, evaluated the use of isomalto-oligosaccharide (IMO) to cover liposomes and protect them from aggregation or hydrolytic degradation. Indeed, IMO allowed them to obtain a liposome of around 250 nm without changing before and after freeze drying. Moreover, the good values of oral bioavailability of bioflavonoids underline the ability of IMO to protect the system from enzymatic and acidic damage [69]. Other excipients listed in the Novel Food Directive No. 2017/2470 are the disaccharides trehalose and cellobiose, well-known cryoprotectants used for preventing the collapse of the vesicles during the freeze drying process.

A different approach used by Wu et al. was the use of chitosan from *A. bisporus*, a wild mushroom from China, to coat liposomes to increase their permanence in the gastrointestinal tract and facilitate mucous penetration. In their study, the authors reported the ability of chitosan from the fungal source to create a corona around liposome with a lower size than commercial chitosan. The stability and the encapsulation efficiency capacity were not impaired by the origin of chitosan [13].

Overall liposomes from compounds derived from Novel Foods originate from the hydration of thin lipid films or lipid cakes, causing them to expand. During agitation, the hydrated lipid sheets detach and naturally reseal themselves, leading to the creation of large liposomes that necessitate the input of energy, typically in the form of sonic energy (sonication) or mechanical energy (extrusion) for reducing size [39,40,41,42,67,68].

In recent years, as well as liposomes, SLNs have been often used for the transport of nucleic acids, such as small-interfering RNAs or messenger RNAs. In this context, the endosomal escape and further release of RNA in the cytosol, after cellular uptake, is the critical point for a successful delivery. The use of ionizable lipids that perturb the endosome can help the escape; however, recent studies have shown that the addition of cholesterol on the NP surface can improve membrane fusion. This capability is conferred by the steroidal structure of cholesterol. Patel et al. created a collection of SLNs by using a microfluidic preparation technique with different phytosterols to test their activity in gene delivery. β-sitosterol appears to be the more promising phytosterol in terms of transfection and modification of SLN morphology, which allows for a more fragile structure by promoting fusion with the endosomal membrane [38].

The second generation of SLNs is represented by the NLCs that are composed of incompatible solid and liquid lipids. The irregular matrix improves the loading of lipophilic active ingredients and the long-term colloidal stability compared to SLNs. Overall, NLCs were investigated in depth for the oral and topic delivery of lipophilic compounds. For the production of NLCs, the Novel Food *Schizochytrium* oil was recently proposed. *Schizochytrium* sp. is a type of microalgae (diatoms) able to store lipids into a droplet. This particular diatom produces an oil that contains polyunsaturated fatty acids, eicosapentaenoic acid, and docosahexaenoic acid.

### 2.3. Extracellular Vesicles

Extracellular vesicles (EVs) are lipid membranes in a nanosized range that can be produced by different cell types. In order to promote the use of sustainable material, the natural metabolites of microalgae, such as nanoalgosomes, a type of EVs secreted with high yields by microalgae, isolated by a tangential flow filtration technique, have been proposed as a new delivery system. The marine chlorophyte *Tetraselmis chuii* and its metabolites are intended as Novel Food [51]. Picciotto et al. [50] and Adamo et al. [51] worked with this type of microalgae to obtain and characterize the nanoalgosomes from the cultivation medium. These research groups have developed a method to separate the nanoalgosomes from the culture medium, thus obtaining EVs with a size of about 100 nm. Tests on cell lines and on *C. elegans*, as an invertebrate model, demonstrated the ability of these EVs to be internalized using epithelial and intestinal cells with the future possibility of being used as natural, sustainable, and ethically accepted delivery systems.

### 2.4. Nanoemulsion Systems

The emulsions are a mixture composed of two different phases that are normally immiscible. With the addition of surfactants or particular techniques, a delivery system can be produced. In particular, nanoemulsion is a colloidal suspension that contains oil and surfactants to obtain a stable formulation and is transparent in appearance with a droplet size less than 200 nm, which is usually obtained using a high-speed disperser.

Different types of oils can be used for the production of nanoemulsion systems. Concerning the Novel Food list, chia seeds contain the highest amount of ɑ-linolenic acid (ALA), an essential fatty acid precursor of omega-3 and phospholipids. The use of ALA for the preparation of a nanoemulsion was investigated by Kuznetcova et al., who obtained a stable formulation of about 100 nm with a negative surface charge [11]. Moreover, sacha inchi oil was exploited by Echeverri et al. to prepare a nanoemulsion with good stability over time, unlike the coarse emulsion [46].

To solve the stability problem, a Pickering emulsion was studied in recent years. This type of emulsion is characterized by the presence of solid particles typically made of silica or clay, even if the use of natural biopolymers can represent an innovative application of Novel Food in this field.

Proteins from potatoes, rich in different amino acids and modified with chitosan, were studied as a new surfactant to improve the stability and antioxidant properties of this emulsion. Hu and Zhang used this combination of compounds to obtain an emulsion with a size of less than 1.5 μm that was stable for up to 28 days in different storage conditions [70].

Another study used chitosan to cover the phytoglycogen surface to obtain an amphiphilic mixture usable to stabilize a Pickering emulsion containing β-carotene. This emulsion with a size of about 22 μm was stable for 6 months and was able to load and protect β-carotene from light and temperature [33].

### 2.5. Protein-Based Nanoparticles

Proteins or peptides are bioactive molecules that are gaining importance in the drug delivery field. As a matter of fact, they are attractive alternatives to synthetic polymers. Protein nanoparticles can be obtained with chemical (emulsion or complex coacervation), physical (spray drying), or self-assembly methods [71]. The advantages of using these biomacromolecules in nanoparticle formulations rely on their high safety, biocompatibility, and biodegradability. Moreover, the amphiphilic properties make the protein highly water-soluble. Therefore, the production of protein nanoparticles does not require the employment of any toxic chemical or organic solvents.

Proteins can be extracted from animal or plant sources. Those extracted from plants are gaining interest for their low allergenicity and sustainability, and are being extracted from agri-food wastes in most cases [72].

Coagulated potato, rapeseed, and mung bean proteins are listed in the Novel Food Directive No. 2017/2470. These natural peptides facilitate the endosomal escape and the release of the active compound in the cytosol, and were thus proposed as excipients for formulating nanoparticles suitable for a magnitude of applications [45].

Edelman et al. used potato proteins to load astaxanthin, a carotenoid, obtaining a particle with a size less than 400 nm for oral administration [17]. Other authors, after rapeseed extraction, used the protein to encapsulate β-carotene or, after cathepsin-B hydrolyzation, to obtain peptides for encapsulating doxorubicin [18]. These nanoparticles were obtained due to the self-assembling features of the protein.

Conversely, mung bean proteins have been exploited as an excipient for the coacervation process in combination with polysaccharides. A coacervate system consists of two polyelectrolyte molecules (e.g., protein and polysaccharide) with opposite charges capable of creating an electrostatic interaction. Mung bean protein was observed to be rich in basic amino acids, such as arginine and histidine, that can interact with negatively charged carboxylic groups present in polysaccharides [73]. Meiguni et al. extracted this protein from mung beans with a yield of 82%, and, after an optimization process, they obtained coacervates with succinylated chitosan able to embed curcumin with an encapsulation efficiency of about 90% and good protection from light [30]. The protection of the cargo from light makes these particles suitable for the loading of several light-sensitive bioactive compounds, such as carotenoids. Mung bean proteins were also exploited by Koralegedara and co-workers. Taking advantage of the cationic feature of the proteins, the authors studied the formation of coacervate using different legume proteins, among which were mung bean proteins and carrageenans. The obtained system had a size of around 130 nm with a negative surface charge and was stable in the pH range of 4.0–7.0, resulting in a promising vehicle for the transport of active molecules [29].

From the Fabaceae family, alfalfa or lucerne (Medicago sativa) leaves were proposed as a new source of proteins for the production of protein-based nanoparticles. These proteins have demonstrated good emulsifying, gelling, and film-forming features [74]. Hadidi et al. developed nanosized protein particles via nano-precipitation for loading hemp essential oil, a by-product of Cannabis sativa processes. The authors proved that alfalfa proteins are a promising tool capable of stabilizing, thermally protecting, and increasing the antioxidant activity of the essential oil [28].

Finally, in January 2023, pea and rice protein fermented by Lentinula edodes mycelia were included in the Novel Food list. These proteins might represent novel sources of proteins with similar chemical features.

Concerning animal sources, eggshell membranes (ESMs) are naturally occurring protein-based thin layers with a thickness of ca. 50–70 μm found between the albumen and shell of eggs which protect against bacterial infections [61]. Although the ESM is classified as a waste of pasteurized liquid in the egg industry, ESMs exhibit microstructural architectures of entangled microfibers rich in 80–85% proteins; 10% collagen (collagen types I, V, and X); and 70–75% keratin, glycoproteins, fibrillin, and proteoglycans [15]. The Novel Food Directive No. 2017/2470 includes the hydrolysate of egg membranes produced with hydro-mechanical separation to obtain egg membranes that undergo a patented solubilization. Several studies report its successful application as a component of biocompatible delivery systems highlighting the therapeutic potential of nanosystems based on EMSs [14,16]. Other important sources of proteins are edible insects. Four edible insects have been approved as Novel Foods so far: *Tenebrio molitor* (mealworm), *Locusta migratoria* (migratory locust), *Acheta domesticus* (house cricket), and the larva from *Alphitobius diaperinus* (lesser mealworm). Notwithstanding the low acceptance of the Western countries of these food matrices, edible insects are gaining attraction as a potential inexhaustible source of edible proteins. Indeed, the farming of edible insects is fast and environmentally friendly. The high water solubility and the strong negative charge of these proteins make them suitable for coacervate production with positively charged polysaccharides. Okagu et al. used non-cuticular protein from *Tenebrio molitor* along with chitosan for loading curcumin [49]. Finally, the successful delivery of curcumin [45] was achieved by using phytoglycogen in combination with casein, a milk protein, following a carboxymethylation step for obtaining polyelectrolyte complexes of 50/100 nm [35].

## 3. Compound Derived from Novel Foods Embedded into Nanocarriers

The European list of Novel Foods in Regulation No. 2017/2470 covers foods traditionally used in non-EU countries as nutritional or medical sources, new substances to be used in foods, food from new sources, as well as new ways and technologies for producing food.

Consequently, numerous Novel Foods listed in the Directive are well-known substances commonly employed in the food industry but obtained via a new industrial process, for which applicants requested authorization because it was more convenient or eco-friendly, such as the processes from agri-food wastes. The European Commission authorizes the use of a Novel Food at the moment in which its consumption is not nutritionally disadvantageous. As an example, the stilbene resveratrol extracted from Japanese knotweed (*Fallopia japonica*) has been largely used in food supplements before 1997. The *trans* isomer of resveratrol obtained from microbial sources or via synthetic processes is considered a Novel Food. Also, epigallocatechin-3-gallate can be found in the Novel Food list as a purified extract from green tea (*Camellia sinensis*) leaves, even though other green tea extracts have been used prior to 1997. Nanotechnology can be considered an important platform for favoring and increasing the beneficial effects of bioactive compounds.

Table 2 summarizes an almost exhaustive list of Novel Foods reported in Directive No. 2017/2470 and embedded for different purposes in nanodelivery systems, highlighting the most important outcomes and advantages of the “nano” strategy application. Since the Novel Foods produced via non-traditional methods exhibit the same nutritional and beneficial features as the traditional ones, studies developed using traditional sources were thoroughly assessed.

Starting from resveratrol and epigallocatechin-3-gallate, their effects on health might be impaired by their low bioavailability or high instability in the physiological environment. The encapsulation of resveratrol and epigallocatechin-3-gallate via different nanocarriers was investigated in depth to increase their beneficial activity, such as their antioxidant, antimicrobial, and anti-aging effects, as recently reviewed by Bohara et al. and Sahadevan et al., respectively [75,76].

The same beneficial activities were also reported for other phytochemicals belonging to phenolic acid, flavonoid, lignan, stilbene, alkaloid, and anthraquinone classes; however, the bioavailability of phytochemicals is quite low after oral intake. Indeed, these bioactive compounds are easily metabolized at the intestinal level by digestive enzymes and microbiota and are degraded because of the low-pH stomach environment [77]. From the Plantae kingdom, taxifolin-rich extract from Dahurian Larch (*Larix gmelinii*) wood, *Magnolia officinalis* bark, Korean angelica (*Angelica gigas*) roots, noni or beach mulberry (*Morinda citrifolia*), and blue honeysuckle (*Lonicera caerulea*) fruits were proposed as new sources of nutraceuticals in Europe by the Novel Food Directive No. 2017/2470. These plants were exploited for centuries in traditional medicine because of the beneficial properties of their phytochemicals (taxifolin, magnolol, pyranocoumarins, anthraquinones, and anthocyanins, respectively), and innovative delivery strategies were recently proposed to enhance their antioxidant, antimicrobial, and anti-cancer activities [78,79,80]. Differently, *Echinacea angustifolia* root and cranberry (*Vaccinium macrocarpon*) fruit extracts have been widely employed as supportive treatments (in dietary supplements) for the prevention of respiratory and urinary tract infections, respectively. These supplements have been authorized even if extracted via innovative methods on the basis of the Novel Food regulation. Recently, the encapsulation of whole extracts in liposomes, niosomes, or polymeric nanoparticles was proposed to accentuate their activities.

Another noteworthy class of plant-derived nutraceuticals is represented by phytosterols. As mentioned before, phytosterols display a similar chemical structure to cholesterol. This feature confers to the lipid-lowering effects of phytosterols. On the other hand, the steroidal structure impairs their bioavailability and bioactivity. The same *Achilles* heel has been noticed for carotenoids and liposoluble vitamins together with a marked photo instability. The poor water solubility and chemical degradation have been easily counteracted by several nanodelivery strategies. The carotenoids lycopene and zeaxanthin are inserted in the food additive list (E160 and E161h) and have been authorized in food supplements for decades for their antioxidant properties. The European Commission allowed the employment of these nutraceuticals obtained via synthetic or microbial processes. Additionally, the production of lycopene via extraction from foods (tomato peel and oleogum resin, and red guava) and mold (*Blakeslea trispora*) was also granted.

From the kingdom Fungi, the authorization of *Antrodia cinnamomea* as a novel active ingredient in food supplements should be mentioned. *Antrodia cinnamomea* was exploited in the traditional medicine of Eastern Asia countries because of the high content of triterpenoid with strong anti-diabetic, anti-inflammatory, and neuroprotective effects. Kong et al. and Menon et al. proposed polymeric nanoparticles and cyclodextrin inclusion complexes to increase the bioavailability of the hydrophobic nutraceuticals.

From the kingdom Animalia, eggshell membrane proteins, Antarctic krill (*Euphausia superba*) oil, and bovine lactoferrin are cited in the Novel Food list as new sources of foods. The eggshell membrane proteins display an important biological activity in addition to interesting delivery properties. Mucoadhesive polymeric nanoparticles were demonstrated to increase the local delivery of eggshell membrane proteins, emphasizing their antioxidant and anti-inflammatory properties and simultaneously preventing their intestinal degradation. Antarctic krill is rich in omega-3 fatty acids (docosahexaenoic and eicosapentaenoic acids) and the carotenoid astaxanthin. These nutraceuticals are active in reducing blood lipid and sugar levels, and their encapsulation in lipid-based nanocarriers demonstrated an ability to efficaciously protect them from light and thermal oxidation. Additionally, astaxanthin is obtained from *Haematococcus pluvialis* algae, which is the most important source of this carotenoid authorized by the Novel Food Directive No. 2017/2470. Abdol Wahab and co-workers recently reported the state of the art in the oral delivery of this carotenoid [81]. Bovine lactoferrin is a protein that naturally occurs in cow’s milk with numerous beneficial activities, such as antimicrobial, anti-cancer, and anti-inflammatory effects [82]. These beneficial properties can be enhanced with nanodelivery carriers capable of enhancing bovine lactoferrin permeability and bioavailability.

Finally, iron hydroxide adipate tartrate (IHAT) is the sole engineered colloidal material reported in the Novel Food list. IHAT is a nutritional source of iron used in food supplements for treating iron deficiencies, such as anemia. In IHAT, the iron oxo-hydroxide is embedded into a corona of tartrate. The dispersion is aided with adipic and tartaric acids. The engineered material in powder form is insoluble in the gastrointestinal tract; however, it is efficiently internalized using enterocytes as whole nanoparticles with endocytosis.

**Table 2 pharmaceutics-15-02614-t002:** Novel Food-derived nutraceuticals embedded into nanoparticles for oral delivery.

Novel Food	Nutraceutical Compounds	Activity	Type of Nanoparticle	Results	Reference
*Angelica gigas* Nakai (AGN) dried root	Pyranocoumarins (e.g., decursin and decursinol angelate)	Antioxidant	HPMC as biopolymer + acetic acid as plasticizer for final solid formulation	Better extraction of phenolic compounds and flavonoids from AGN using HPMC and plasticizer	[83]
Astaxanthin-rich oleoresin from *Haematococcus pluvialis* algae	Astaxanthin	Antioxidant, anti-cancer, and cardiovascular disease prevention agent	Nanoemulsions, liposomes, SLNs, chitosan-based and PLGA-based nanoparticles	Improved bioavailability	[81]
Antarctic krill oil from *Euphausia Superba*	Omega-3 fatty acids (mainly docosahexaenoic and eicosapentaenoic acids) and astaxanthin	Reduces blood lipid, sugar levels, and the risk of atherosclerosis, slows down nerve aging, relieves the symptoms of depression, and anti-inflammatory	NLC	Protection of polyunsaturated fatty acids from oxidation, light, and temperature	[84]
*Antrodia camphorata* (or *cinnamonea*)	Triterpenoids	Anti-diabetic	Silica/chitosan	Demonstrate anti-diabetic properties on rat model and protection of testicular dysfunction	[85]
Bovine lactoferrin	Bovine lactoferrin	Anti-viral against SARS-CoV-2	Phosphatidil coline liposome	Reduction of the infection of about 80% on the lung cell line, and a higher anti-viral effect than non-liposomal lactoferrin. Protection from gastrointestinal conditions after oral administration	[86]
Cranberry extract powder from *Vaccinium macrocarpon*	Polyphenols (mainly proanthocyanidins)	Antioxidant	Liposomes containing bile salt in the double layer to inhibit enzymes in the gastrointestinal tract	Protection of the liver by the reduction of the level of antioxidant enzymes	[87]
		Antimicrobial	Nanoemulsion with green tea catechins	Synergic effect against *E. coli* for urinary infection treatment	[88]
		Antioxidant	Chitosan/carrageenan NPs	Prevention of their degradation in the digestive medium	[89]
		Antimicrobial	Chitosan NPs	Interaction with bacteria reducing adherence to intestinal tissue	[90]
*Echinacea angustifolia* extract from cell cultures	Echinacoside and polyphenols	Antimicrobial	Alginate/chitosan nanoparticles	The encapsulated extract displayed a higher biofilm inhibition and up to a 32-fold lower MIC compared to the free extract against *Staphylococcus aureus*	[91]
			Niosome	The encapsulated extract showed up to a 16-fold greater antibacterial activity against *Klebsiella pneumoniae* compared to the free extract	[92]
Egg membrane hydrolysate proteins	Collagen, elastin, glycosaminoglycans	Antioxidant, anti-inflammatory against IL-8 in intestinal epithelial cells	pH sensitive chitosan/fucoidan NPs	Protection from intestinal degradation, increased antioxidant activity, and mucoadhesion that increase local delivery	[22]
Epigallocatechin-3-gallate as a purified extract from green tea leaves (*Camellia sinensis*)	Epigallocatechin-3-gallate	Antioxidant, anti-inflammatory	Chitosan-based NPs	Controlled release in the intestinal environment and intestinal protection of epigallocatechin-3-gallate (preserved antioxidant activity)	[93]
Iron hydroxide adipate tartrate		Iron deficiency such as anemia	NPs of iron hydroxide adipate tartrate is insoluble in GI tract	Slowly ferrous iron release avoiding ROS production	[94,95,96]
*Lonicera caerulea* berries	Anthocyanins	Antioxidant and antimicrobial	PLGA and carboxymethyl chitosan NPs	Improved therapeutic efficiency	[80]
Lycopene (LYC) produced with a synthetic process or extracted from *Blakeslea trispora* (red guava), tomato peels, or oleoresin from tomatoes	-	Antioxidant	Polyelectrolyte complexes with sodium caseinate and TLH-3, an acidic polysaccharide	Protection from oxidation and controlled release in the gastrointestinal tract	[97]
	-	Antioxidant, anti-inflammatory	Liposome with lecithin and cholesterol	Increased level of LYC in serum and brain compared with free LYC. Reduction in ROS and inflammation	[98]
		Antioxidant, anti-tumoral (prostate cancer)	Self-emulsifying system with coconut oil and sorbitan monostearate	Confirm the antioxidant activity without side effects. Ability to reach prostate tissue	[99]
	-	Antioxidant for liver disease	Chitosan-based NPs	Hepatoprotection with reduction in IL and TNF-alpha	[100]
	-	Antioxidant	Catechin NPs coated with chitosan	Gastric protection and higher plasma concentration	[101]
	-	Antioxidant, anti-tumoral (breast cancer)	Whey protein isolate NPs	High plasma concentration after oral administration	[102]
Magnolia bark extract (*Magnolia officinalis*)	Magnolol and honokyol	Antioxidant, anti-inflammatory, anti-cancer, antidepressant, and for the treatment of ulcers	Mixed micelles and nanosuspensions	Increased permeation of magnolol in Caco-2 cell lines and in vivo gastrointestinal absorption	[103]
			Amorphous solid dispersion using HPMC acetate succinate	Increased bioavailability with antioxidant activity and gut protection	[104]
			Protein NPs into chitosan/alginate hydrogel MPs	Release in the colon; uptake and anti-inflammatory effect on epithelial and macrophage colon cells	[105]
			Micelles with Pluronic F127 and L61 or with copolymers (Soluplus, Solutol, and D-alpha-toco-pheryl PEG 1000 succinate)	pH-dependent release in the intestine; bioavailability 3-fold greater than raw product	[106,107]
Noni fruit juice Noni fruit juice powder Noni fruit puree and concentrate (*Morinda citrifolia*)	Anthraquinones (damnacanthal) and polyphenols	Anti-cancer	PLGA-PEG nanocapsules	Higher activity in cell growth inhibition, compared to non-encapsulated damnacanthal	Reviewed by [79]
Phytosterols/phytostanols		Anti-cancer	SLNs loading several phytosterols using different glycerides	Increased solubility (bioavailability) and better hypocaloric properties	[108]
		Anti-cancer (breast cancer)	Chitosan/alginate NP functionalized with folate for breast cancer-targeting delivering β-sitosterol	Protection to enzymes and hydrolysis; prolonged release; good permeation across intestinal cells; and high toxicity against cancer cells	[109,110]
		LDL cholesterol- lowering properties	Nanoporous starch aerogel impregnated with phytosterol from soybeans	Increased bioavailability from 3 to 35%; reduction of crystallinity	[111,112]
		Cholesterol-lowering properties	Soy protein vehicles delivering several phytosterols	Reduction of cholesterol level and better bioavailability	[113]
	-	Cholesterol-lowering properties	Sodium caseinate/pectin coacervate	Protection in the gastrointestinal tract; hydrophobic bond sodium caseinate reduce crystallinity and increase bioaccessibility after digestion	[114]
		Cholesterol-lowering properties	Soybean protein/pectin coacervate delivering stigmasterols	Stability at different pH values with an increased stability to the stomach environment; release in intestinal fluids	[115]
		Cholesterol-lowering properties	Zein/pectin coacervate delivering stigmasterols	Pectin creates a gel around zein/phytosterol	[116]
Taxifolin-rich extract from the wood of Dahurian Larch (*Larix gmelinii*)	Taxifolin	Antioxidant	Micelles, liposomes, polymeric NPs, and hybrid systems	Improved stability, permeability, and systemic availability of quercitin, a taxifolin analogue	Reviewed by [117]
*Trans*-resveratrol produced with a synthetic process		Antimicrobial, antioxidant, anti-aging	SLNs, liposomes, dendrimers, polymeric NPs	Increased bioavailability and permeability. Increased concentration in the brain compared to free resveratrol	Reviewed by [75]
Vitamin K2 (menaquinone) produced with a synthetic or microbiological process	-	Fat-soluble vitamin deficiencies in pancreatic-insufficient CF patients	Liposomes containing a blend of vitamins + vit. K2	Increased level of other vitamins in plasma in comparison with the same supplements without vitamin K2 (trial in cystic fibrosis patients)	[118]
	-	Bone support, promotes heart health and helps boost immunity	Liposomes loaded with vit. D3/K2 and coated with chitosan with mucoadhesive- properties	High encapsulation and controlled release in situ	[119]
Zeaxanthin produced with synthesis	Zeaxanthin	Antioxidant, cardiovascular disease prevention agent	Liposomes, nanoemulsions, polymeric NPs, and polymer–lipid hybrid NPs	Increased bioavailability and stability of lutein, the isomer of zeaxanthin	Reviewed by [120]

## 4. Nanoceutical Application in the Food Sector: Safety Issues and Regulations

Nanocarriers loaded with bioactive compounds or produced using materials derived from foods for influencing human health or condition can find application in different fields, such as pharmaceutical, health, cosmetic, and food. In this latter case, nanoceuticals can be designed for fortifying foods or obtaining dietary supplements to increase the total nutrient profile of a diet. Besides the advantages and benefits, the employment of nanoparticles has raised concerns regarding the safety of their intake, which has not been fully elucidated. Even though a material is considered safe, its physiochemical properties at a nanoscale are completely different. The oral intake of nanoparticles might alter the normal functions of the gastrointestinal tract [121] and gut microbiota [122] or increase the risk of accumulation within tissues and cells depending on their dimensions, shape, and surface charge, as discussed in depth by other authors [123,124,125]. Moreover, nanotechnology can enhance the oral bioavailability of hydrophobic bioactive compounds causing adverse consequences or health problems in some cases [124].

The first advantage of using Novel Foods as materials for obtaining ingredients for the production of nanoceuticals or bioactive compounds is related to the reuse of agri-food wastes. In March 2020, the European Commission adopted the “new circular economy action plan” within the “European green deal” for more sustainable growth. Thus, the recovery of compounds from waste (e.g., lycopene from tomato peel or eggshell membrane proteins) is fundamental for achieving the goal set out by the European Union. Another advantage can be represented by the employment of new sustainable sources in light of the dramatic foreseen scenario alerted by the Food and Agriculture Organization, as mentioned in the introduction. The exploitation of algae, microorganisms, or insects are a few examples [126].

In Europe, the application of nanotechnology in food products (e.g., food supplements and fortified foods) is regulated by Directive No. 2015/2283 concerning Novel Foods, which reports that ‘engineered nanomaterial’ refers to “any intentionally produced material that has one or more dimensions of the order of 100 nm or less or that is composed of discrete functional parts, either internally or at the surface, many of which have one or more dimensions of the order of 100 nm or less, including structures, agglomerates or aggregates, which may have a size above the order of 100 nm but retain properties that are characteristic of the nanoscale”. When the definition of a nanomaterial is fulfilled, the authorization process described above is no longer sufficient. Indeed, in this case, EFSA authorization is granted only after a careful evaluation of toxicological hazards that might require in vitro and in vivo studies.

EFSA has recently drafted two guides on the risk assessment of nanomaterials that outline (i) the technical requirements for establishing the presence of small particles and (ii) the scientific risk assessment and the necessary testing for evaluating the safety to protect consumers [127,128]. The regulatory safety assessment of nanoparticles in foods has been recently discussed in depth by Schoonjans and co-workers [129].

Briefly, the EFSA Scientific Committee had considered scientific studies that provide a relevant understanding of hazard characterization, exposure assessment, and physico-chemical properties of nanomaterials. The first step for the risk assessment is the physico-chemical characterization of the material. In particular, particle size is the most important feature. As a matter of fact, the scientific literature underlined that particles up to 250 nm have a high chance of translocation from the gastrointestinal tract to the tissues. Thus, the Scientific Committee established that particles with a size equal to or larger than 500 nm with less than 10% of particles with a smaller size (number-based particles) are not engineered nanomaterials and can be approved with a conventional risk assessment. If these particles are composed of ingredients present in the lists of Novel Foods (Regulation (EC) No. 2015/2283), substances for the fortification of foods (Regulation (EC) No. 1925/2006), foods for specific groups (Regulation (EU) No. 609/2013), or food additives (Regulation (EC) No. 1333/2008), they are considered as safe and can be used without any risk assessments. On the contrary, particles with a dimension equal to or smaller than 500 nm with more than 10% of particles with a smaller size need a nano-specific risk assessment. The Scientific Committee selected electron microscopy as the recommended technique for size screening. If the electron microscopy measurement is not feasible, other methods are indicated (e.g., scattering techniques). The data have to be presented as a number size distribution and are estimated through Feret’s diameter. Feret’s diameter is a commonly used measure in microscopy that considers both the maximum and the minimum diameter of an irregularly shaped particle. Small particles are not considered ‘engineered nanomaterials’ when they lose their particulate nature due to physical, chemical, or biological processes, or solubilization in the gastrointestinal tract, because their risk is expected to be similar to the conventional bulk material.

According to EFSA guidelines, if the material displays nanomaterial features after physico-chemical characterization, the second step of the nano-specific risk assessment involves in vitro digestion by reviewing the existing information or generating new data, including genotoxicity and cell toxicity. Subsequently, if the material is persistent or there are indications of toxicity, step 3 comprises in vivo testing for determining pharmacokinetic profiles and the histopathology of gastrointestinal sites and organs [127,128].

Definitely, many researchers have pointed out the complementary activation, cytokine release, and other immunological and hematological reactions, of the nanoparticles, leading to compromised safety and efficacy. According to a recent review, the activation of the immune system, which carries inflammatory reactions, is almost exclusively induced by inorganic nanoparticles, mainly carbon-based or metal-based nanoparticles, particularly containing titanium dioxide, gold, silica, and graphene oxide–based nanomaterials [130]. It is clear that the nature of the lipids and polymers of the organic nanoparticles strongly affects their safety profile. It is widely accepted that biodegradable and biocompatible natural organic compounds (principally polysaccharides and proteins) that are Generally Recognized as Safe (GRAS) have become attractive players for the development of safe and powerful nanoparticles based on natural constituents [131,132].

## 5. Conclusions

Nanosized delivery systems represent an emerging sector in health, pharmaceutical, cosmetic, and food areas with great potential, progressing fast and using a variety of materials, coatings, and functions. The aim of this paper was to provide a comprehensive review of the literature, which combines the nanodelivery systems and phytoconstituents derived from Novel Foods, but is also evidenced as Novel Food and could represent an innovative source of materials for the formulation of nanodelivery systems. The goals of the reviewed studies principally comprised increased solubility, higher stability, and improved bioavailability of the loaded phytoconstituents.

Undoubtedly, nanoparticles naturally occur in food because numerous food and feed constituents consist of inherent proteins, carbohydrates, and fats, which span a range of sizes from large biopolymers (macromolecules) down to the nanoscale. The primary concern, however, revolves around whether synthetic nanoparticles and other nanomaterials may pose any potential risks. EFSA has recently produced guidelines, which can deeply help the development of robust, integrated, and acceptable testing methods; however, a harmonized and detailed approach is not yet available to assess their adequate safety profiles. Critical points are probably related to the biodegradation of the nanocarriers and their biocompatibility—mainly related to their nature (GRAS nanosystems) and their production—which should avoid the use of hazardous chemical reagents and also create systems (nanovectors) mimicking natural exosomes, which cannot be recognized by our body as harmful vectors that induce immunological reactions or other adverse events.

## Figures and Tables

**Figure 1 pharmaceutics-15-02614-f001:**
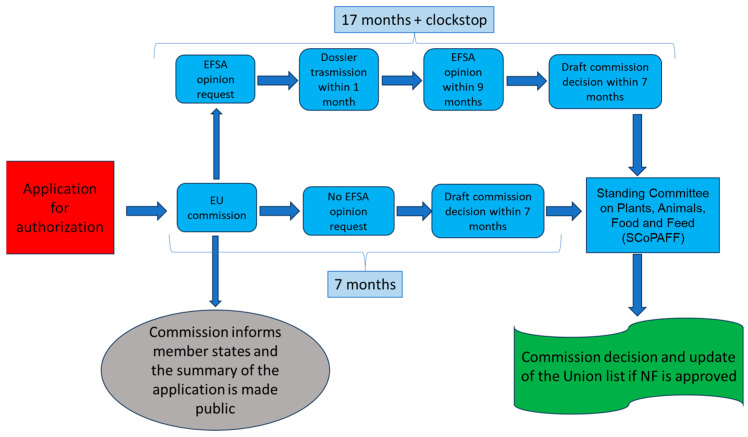
Scheme of the authorization process of Novel Foods in Europe.

**Figure 2 pharmaceutics-15-02614-f002:**
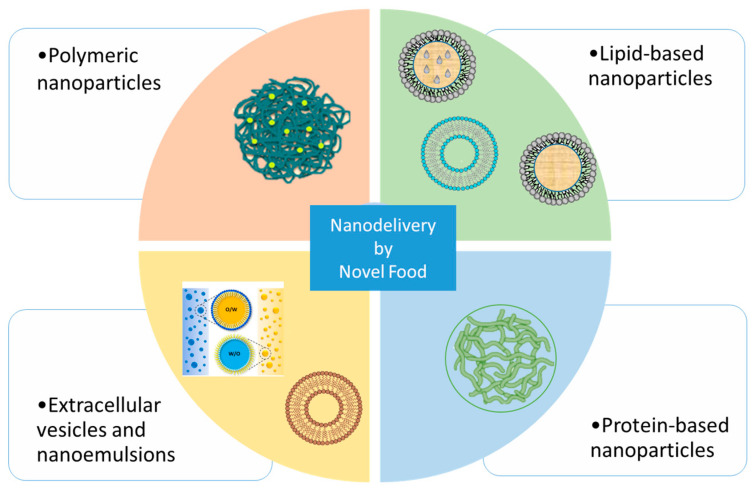
Novel foods as bioactive compounds or sources of materials for nanocarrier development.

**Table 1 pharmaceutics-15-02614-t001:** Materials listed in the Novel Food catalog and used for producing nanoparticles.

Material(s) from Novel Foods	Type of Carrier	References
*Astragalus membranaceus* root extract	Polysaccharide nanoparticles	[8,9]
Cellobiose	Cryoprotectant for liposomes	[10]
Chia seed oil from *Salvia hispanica* L.	Liposomes and nanoemulsions	[11]
Chitosan extracted from fungi (*Aspergillus niger*; *Agaricus bisporus*)	Chitosan nanoparticles	[8,12,13]
Chondroitin sulphate (synthetic)	Polysaccharide nanoparticles	[14,15,16]
Coagulated potato proteins	Protein-based nanoparticles	[17,18]
Dextran from *Leuconostoc mesenteroides*	Polysaccharide nanoparticles	Reviewed by [19]
*Digitaria exilis*	Polysaccharide nanoparticles	[20]
Eggshell membrane protein hydrolysate	Protein-based nanoparticles	[21,22,23]
Fucoidan extracted from the seaweed *Fucus vesiculosus* and *Undaria pinnatifida*	Polysaccharide nanoparticles	[24,25,26]
Guar gum	Polysaccharide nanoparticles	[27]
Lucerne leaf extract from Medicago sativa	Protein-based nanoparticles	[28]
Mung bean seed proteins from *Vigna radiata*	Protein-based nanoparticles	[29,30]
*Panax notoginseng root extract*	Polysaccharide nanoparticles	[31]
Phytoglycogen	Polysaccharide nanoparticles Polyelectrolyte complex	[32,33,34,35,36,37]
Phytosterols	Solid lipid nanoparticles Liposomes	[38,39]
Phospholipids from egg yolk	Liposomes	[40,41,42]
Phosphatidylserine from soya and fish phospholipids	Liposomes	[43,44]
Rapeseed protein from *Brassica napus* L. and *Brassica rapa* L.	Protein-based nanoparticles	[18,45]
Sacha inchi seed oil from *Plukenetia volubilis*	Nanoemulsions	[46]
*Schizochytrium* sp. oil	Nanostructure lipid nanoparticles	[47]
Sugar cane fiber	Polysaccharide nanoparticles	[48]
*Tenebrio molitor* L.	Protein-based nanoparticles	[49]
*Tetraselmis chuii* microalgae	Extracellular vesicles	[50,51]
Trehalose	Cryoprotectant for liposomes	[10]
Yeast β-glucan	Polysaccharide nanoparticles	[52,53]

## Data Availability

Data are contained within the article.

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
