# Peer review of "Combination of Nanodelivery Systems and Constituents Derived from Novel Foods: A Comprehensive Review"

_pharmaceutics, 2023, doi:10.3390/pharmaceutics15112614_

Round 1

Reviewer 1 Report

Comments and Suggestions for Authors

The submitted paper is an interesting overview on Novel Foods (based on natural compounds) achievable from application of nanotechnologies.

My opinion is positive, the paper is well organized and rich of information. The References are recent. I just have one observation (and suggestion). The manuscript describes different types of nanoplatforms but their preparation technologies are not described (the recent ones, the most applied...).

I understand that the paper is focused on nanocarriers with natural components for novel foods but short presentations (in a separate section or at the end of the sections 2.1 Polymeric nanoparticles, 2.2 Lipid-based nanoparticles and so on...) on techniques and processes would be useful to the Readers to know how cited nanocarriers (or part of them) were produced. This information will give, in my opionion, an effective "comprehensive review" characteristic to the submitted paper.

Author Response

We would like to thank the referee for the positive evaluation of the paper and for the valuable suggestion aimed to enhance the quality and readability of the manuscript. As recommended, in the sections of the individual preparations, a short description of the main methods used in the preparation of nanosystems has been added (L335-340, L 392- 397, L404, L422, L436 of revised version), except for the section “2.5. Protein-based nanoparticles” because here,  formulation methods were already indicated. (L460 revised version).

Reviewer 2 Report

Comments and Suggestions for Authors

Comments to the authors:

The work described in the present manuscript is consistent with the scope of the journal.

Authors reviewed the recent advances on the design and characterization of nanodelivery systems based on materials belonging to the Novel Food list, or nanoformulations produced for delivering compounds from Novel Foods. They also discussed the challenges regarding the safety of these nanoparticles. The work is very complete and presents scientific interest, but some major corrections are needed to be performed prior to a possible publication, specifically:

  1. Abstract: Please mention in the abstract that the focus of the review is mainly the state of art of this topic of research in the context of the European Union.
  2. Introduction: Authors must include at least a paragraph describing the scenario in the other regions of the world, particularly some relevant like USA. They must mention if there are relevant differences in the authorization process in comparison to the described in the text and in Figure 1 for other regions.
  3. Figure 3: The white colour of the next is difficult to read and I believe that black colour will be better. Please consider modifying it. Moreover, the figure caption must be correct to refer that the depicted process is the one considered in EU.
  4. Line 85: Replace “which deals with developing new delivery systems” by “which deals with developing, amongst others, new delivery systems”.
  5. Table 1: in the line regarding “Phytosterols”, please replace “Polymeric” by the type of the polymer(s) used, to be in accordance with all other examples of polymeric nanoparticles described in the table.
  6. Lines 182-185: The information described in this paragraph is repeated in lines 363-368. My suggestion is to remove the first paragraph and keep the second one since is more detailed and liposomes are at this point already discussed.
  7. Line 241: The “Directive” is mentioned several times in the manuscript, but authors must clearly specify herein what is the Directive that is being cited.
  8. Line 254: Replace “U.S. Food and Drug Administration” by “U.S. FDA agency”.
  9. Lines 333-336: the work reported in this paragraph (reference 37) seems not relevant for this review since the developed liposomes have 10 micrometer of diameter and thus they cannot be considered as “nano” systems. Please remove this citation.
  10. Lines 573-576: this paragraph must be rewritten since it is not clear if the Bovine lactoferrin was used to produce the nanoparticles or if the nanoparticles were made with another matrix to encapsulate the protein.
  11. Please replace “anti-infective,” by “antimicrobial”.
  12. Table 2: in the line regarding “Bovine lactoferrin”, consider calling the protein as the “nutraceutical compound”. From my understanding, the nanoparticles were made from PC liposomes and then they encapsulated the protein. Also, in the “results”, authors must clarify what is the meaning of “free lactoferrin”.
  13. Table 2: if possible, include limits for horizontal lines since in some of the rows is very difficult to understand what part of the text belong to one reference of to another, for example in the first lines of page 2.
  14. Table 2: anemia is a particular example of iron deficiency, so please rewrite this like for example “Iron deficiency, such as anemia”.
  15. Table 2, page 3: “Self-emulsifying system with coconut oil and sorbitan monostearate (preparation in mild conditions for lycopene stability)”. The text between parenthesis are details and must be deleted.
  16. Table 2, page 3: “Protein NPs into chitosan/alginate hydrogel MPs for ulcerative colitis treatment”. The text “for ulcerative colitis treatment” can be moved to the column “Activity” (or removed).

Author Response

REFEREE 2

The work described in the present manuscript is consistent with the scope of the journal.

Authors reviewed the recent advances on the design and characterization of nanodelivery systems based on materials belonging to the Novel Food list, or nanoformulations produced for delivering compounds from Novel Foods. They also discussed the challenges regarding the safety of these nanoparticles. The work is very complete and presents scientific interest, but some major corrections are needed to be performed prior to a possible publication, specifically:

  1. Abstract: Please mention in the abstract that the focus of the review is mainly the state of art of this topic of research in the context of the European Union.

We specified that the regulation is referred to the European Union.

  1. Introduction: Authors must include at least a paragraph describing the scenario in the other regions of the world, particularly some relevant like USA. They must mention if there are relevant differences in the authorization process in comparison to the described in the text and in Figure 1 for other regions.

A brief description according to the referee suggestion was added in the introduction at lines 59-65 of revised version

  1. Figure 3: The white colour of the next is difficult to read and I believe that black colour will be better. Please consider modifying it. Moreover, the figure caption must be correct to refer that the depicted process is the one considered in EU.

The figure and the caption were modified.

  1. Line 85: Replace “which deals with developing new delivery systems” by “which deals with developing, amongst others, new delivery systems”.

Modified according to the suggestion.

  1. Table 1: in the line regarding “Phytosterols”, please replace “Polymeric” by the type of the polymer(s) used, to be in accordance with all other examples of polymeric nanoparticles described in the table.

We thank the referee. There was an error, “polymeric nanoparticles” was removed.

  1. Lines 182-185: The information described in this paragraph is repeated in lines 363-368. My suggestion is to remove the first paragraph and keep the second one since is more detailed and liposomes are at this point already discussed.

Ok we deleted the repetition as suggested

  1. Line 241: The “Directive” is mentioned several times in the manuscript, but authors must clearly specify herein what is the Directive that is being cited.

The identification number of the directive was added.

  1. Line 254: Replace “U.S. Food and Drug Administration” by “U.S. FDA agency”.

Modified accordingly.

  1. Lines 333-336: the work reported in this paragraph (reference 37) seems not relevant for this review since the developed liposomes have 10 micrometer of diameter and thus they cannot be considered as “nano” systems. Please remove this citation.

This part was removed.

  1. Lines 573-576: this paragraph must be rewritten since it is not clear if the Bovine lactoferrin was used to produce the nanoparticles or if the nanoparticles were made with another matrix to encapsulate the protein.

The paragraph was modified as suggested.

  1. Please replace “anti-infective,” by “antimicrobial”.

Modified accordingly.

  1. Table 2: in the line regarding “Bovine lactoferrin”, consider calling the protein as the “nutraceutical compound”. From my understanding, the nanoparticles were made from PC liposomes and then they encapsulated the protein. Also, in the “results”, authors must clarify what is the meaning of “free lactoferrin”.

We modified the table according to the referee's suggestion by placing "lactoferrin" as a nutraceutical compound. Free lactoferrin means not encapsulated in liposomes. We have clarified this in the text of the table 2.

  1. Table 2: if possible, include limits for horizontal lines since in some of the rows is very difficult to understand what part of the text belong to one reference of to another, for example in the first lines of page 2.

Ok, but we used the template of the journal.

  1. Table 2: anemia is a particular example of iron deficiency, so please rewrite this like for example “Iron deficiency, such as anemia”.

Modified accordingly.

  1. Table 2, page 3: “Self-emulsifying system with coconut oil and sorbitan monostearate (preparation in mild conditions for lycopene stability)”. The text between parenthesis are details and must be deleted.

Modified accordingly.

  1. Table 2, page 3: “Protein NPs into chitosan/alginate hydrogel MPs for ulcerative colitis treatment”. The text “for ulcerative colitis treatment” can be moved to the column “Activity” (or removed).

Modified moving the text in the activity column

Reviewer 3 Report

Comments and Suggestions for Authors

Authors describe modern nano-containers from derived from the material of food processing with additional stress to Novel foods sources. The review contains a lot of data on regulation of novel foods and the examples of application of nanocontainers from food origin.

However, some corrections should be made in the manuscript prior submission.

According to IUPAC terminology, submicron particles (100 nm – 1000 nm) can not be named nanoparticles. Some exceptions are not discussed in this paper. So, information in L85, L130 should be corrected.

L86 “Nanocarriers can be roughly classified as matrix particles or vesicles with a liquid core.” This statement is wrong. Authors are focused on these types of nanocarriers, but there are a lot of different structures of nanocarriers.

General claim: with the current title of the text, one could expect discussion of the novel foods’ sources of the nanocarriers. Nevertheless, the information in the review deals with conventional sources of reagents and conventional food and food wastes: egg yolk, shrimps shells, mushrooms and so on.

In addition, is there any difference in chemical properties of molecules (for example lipids) derived from normal food and “new food”? Does it worth attention? The soybean and egg yolk phosphatidylcholines are on the market for decades. The chapter 4 should be broadened with the discussion on the real importance of novel foods’ sources and benefits (if any) of these sources.

Also, I suggest to add the list of the examples of New foods as supplementary material or appendix section.

Author Response

REFEREE 3

Authors describe modern nano-containers from derived from the material of food processing with additional stress to Novel foods sources. The review contains a lot of data on regulation of novel foods and the examples of application of nanocontainers from food origin.

However, some corrections should be made in the manuscript prior submission.

According to IUPAC terminology, submicron particles (100 nm – 1000 nm) can not be named nanoparticles. Some exceptions are not discussed in this paper. So, information in L85, L130 should be corrected. 

We thank the referee for the clarification. As suggested, the terminology has been corrected as follows:

L 93 (revised version): “An emerging area in the pharmaceutical and medicinal fields is nanotechnology, which deals with developing new delivery systems at the nanometric (1 nm – 100 nm) and micrometric (100 nm – 100 μm) range (Michel Vert, Yoshiharu Doi, Karl-Heinz Hellwich, Michael Hess, Philip Hodge, Przemyslaw Kubisa, Marguerite Rinaudo, and François Schué. Terminology for biorelated polymers and applications (IUPAC Recommendations 2012)*. Pure Appl. Chem., Vol. 84, No. 2, pp. 377–410, 2012 as reference n.5].

L142 (revised version): Polymeric particles (PNp) intended as nano and micro-particles are an important class of drug delivery system for the suitable delivery of bioactive compounds composed of either biodegradable or not biodegradable biopolymers. …

L86 “Nanocarriers can be roughly classified as matrix particles or vesicles with a liquid core.” This statement is wrong. Authors are focused on these types of nanocarriers, but there are a lot of different structures of nanocarriers.

According to the suggestion we corrected the statement as follow

L 96 (revised version): Several nanocarriers have received approval for use as drug delivery vehicles in human applications, exhibiting a wide range of compositions and structures. In particular, this review focuses primarily on the prevalent polymeric and lipid-based nanoparticles

General claim: with the current title of the text, one could expect discussion of the novel foods’ sources of the nanocarriers. Nevertheless, the information in the review deals with conventional sources of reagents and conventional food and food wastes: egg yolk, shrimps shells, mushrooms and so on.

Thank you for this request. The law which describes Novel Food reports: "a food that was not used for human consumption to a significant degree within the Union before the date of entry into force of that regulation, namely 15 May 1997”. According to this definition, there are on the market a great number of natural substances and extracts which are linked to this status even if they have been on the market from more than 20 years, just after 1997. It seems a contradiction but after 20 or more years of marketing they are still "Novel" in the food sector.

In addition, is there any difference in chemical properties of molecules (for example lipids) derived from normal food and “new food”? Does it worth attention? The soybean and egg yolk phosphatidylcholines are on the market for decades. The chapter 4 should be broadened with the discussion on the real importance of novel foods’ sources and benefits (if any) of these sources.

Novel food is not a new food but simply a category of food regulated by the European Union Directive No. 2015/2283, and there is a list available on the internet by the European Commission (https://food.ec.europa.eu/safety/novel-food/authorisations/union-list-novel-foods_en). The possible use of these components of Novel Food simply represents a new opportunity of source, in most of the cases the structure is the same of the other food category or they can be different (es. cis trans isomeritzation of lipids). Chapter 4 is revised according to better explaining this concept:

L635 – 644 (revised version): “The first advantage of using Novel Foods as materials for obtaining ingredients for the production of nanoceutical or bioactive compounds is related to the reuse of agri-food wastes. In March 2020, the European Commission adopted the “new circular economy action plan” within the “European green deal” for a more sustainable growth. Thus, the recovery of compounds from waste (e.g. lycopene from tomato peel or eggshell membrane proteins) is fundamental for achieving the European Union goal. Another advantage can be represented by the employment of new sustainable sources in light of the dramatic foreseen scenario alerted by the Food and Agriculture Organization as mentioned in the introduction. The exploitation of algae, microorganisms, or insects are a few examples.”

Also, I suggest to add the list of the examples of New foods as supplementary material or appendix section.

The website of the list has been added according, so the reader can have a continuous update of the revisions (ref n. 3).

Round 2

Reviewer 2 Report

Comments and Suggestions for Authors

the modifiations clearly improved the manuscript. the paper is now suitable for publication.